# A Cost-Efficient MFCC-Based Fault Detection and Isolation Technology for Electromagnetic Pumps

Ugochukwu Ejike Akpudo  and Jang-Wook Hur *

Department of Mechanical Systems Engineering, Kumoh National Institute of Technology,
61 Daehak-ro (yangho-dong), Gumi, Gyeongbuk 39177, Korea; akpudougo@gmail.com
* Correspondence: hhjw88@kumoh.ac.kr

**Abstract:** Fluid pumps serve critical purposes in hydraulic systems so their failure affects productivity, profitability, safety, etc. The need for proper condition monitoring and health assessment of these pumps cannot be overemphasized and this has resulted in extensive research studies on standard techniques for ensuring optimum fault detection and isolation (FDI) results for these pumps. Interestingly, mechanical vibrational signals reflect operating conditions and by exploring the robust time–frequency-domain feature extraction techniques, the underlying nonlinear characteristics can be captured for reliable fault diagnosis/condition assessment. This study is based on the use of vibrational signals for fault isolation of electromagnetic pumps. From the vibrational signals, Mel frequency cepstral coefficients (MFCCs), the first-order and the second-order differentials were extracted and the salient features selected by a rank-based recursive feature elimination (RFE) of uncorrelated features. The proposed framework was tested and validated on five VSC63A5 electromagnetic pumps at various fault conditions and isolated/classified using the Gaussian kernel SVM (SVM-RBF-RFE). Results show that the proposed feature selection approach is computationally cheaper and significantly improves diagnostics performance. In addition, the proposed framework yields a comparatively better diagnostics results on electromagnetic pumps in comparison with other diagnostics methods, hence a more reliable diagnostics tool for electromagnetic pumps.

**Keywords:** Mel frequency cepstral coefficient; electromagnetic pumps; feature selection; recursive feature elimination; support vector machine

## 1. Introduction

The growing demand for increased productivity, maintainability and reliability have motivated the proliferating research studies on the state-of-the-art condition-based maintenance approach—prognostics and health management (PHM). Consequently, optimized reliability has shown strong dependence on prediction-based data-driven PHM [1]. These data-driven PHM approaches have evident comparative advantages against the traditional model-based methods and are even being more appreciated/patronized for the ease of use, minimal false alarm rate, computational cost efficiency associated with them.

The growth of artificial intelligence (AI), machine learning (ML) and deep learning (DL)-based methods have recently motivated the high discrimination against the traditional statistical model-based approaches for FDI [2]. Theoretically, DL methods are quite popular for high fault detection accuracy; however, issues of interpretabiity, high dependence on excessive parameters, overfitting/underfitting issues, computational cost (and complexity) and the magical defiance from fundamental statistical theory [3] make them practically unreliable for cost-aware industrial applications. On the other hand, though not as accurate/automated as DL methods, most Bayesian ML methods come with benefits ranging from relatively minimal false alarm rate, interpretability and computational cost efficiency on few data. Nevertheless, whether statistical-based, data-driven or hybrid, by monitoring and accurately identifying key fault parameters in systems (or components) via sensors,

downtime can be reduced, productivity increased, costs minimized and predicting the end-of-life (EOL) of these systems (or components) reliably possible [4].

FDI entails condition monitoring of systems, identifying when a fault occurs and recognizing the type of fault and its location in the systems [5]. The FDI technology presented herein is based on extracting time–frequency-domain features from the vibrational signals, a rank-based discriminative feature selection and a classification process. This study proposes the use of vibrational signals for FDI of electromagnetic pumps. These pumps serve a critical function in hydraulic and/or thermodynamic systems—to supply fuel (or other fluids) to a desired location at a desired pressure. For this sole purpose, they are usually operated for as long as required of them; thereby exposing them to diverse failure modes from sources ranging from unfavorable environmental conditions, fatigue, fluid contamination, mechanical/electrical stress, uncertainty, filter failure/clogging, overuse/underuse, etc. [6]. Consequently, the need for real-time health monitoring (and assessment) becomes crucial for accurate diagnosis, reliable prognosis and more profitable decision making.

In the quest for ensuring accurate and cost-efficient vibration-based diagnostics frameworks, diverse filter-based, embedded and wrapper-based feature selection approaches are bound for exploration [7–9]; however, every method comes with pros and cons. These cons can be significantly mitigated by hybrid methods (a combination of two or more techniques). By providing an avenue where the strength of one complements the weakness of the other; indirectly, only optimal (highly discriminant) features can be selected for use by a reliable ML-based classifier. These factors, in addition to current computational cost concerns, have motivated our study. To validate the proposed model's diagnostics performance, we tested the model on our testbed which consists of five VSC63A5 electromagnetic pumps produced by Korea Control Limited. While expanding the contributions of our previous study [10], this paper presents the results for intelligent fault detection of the five VSC63A5 electromagnetic pumps at different operating conditions.

## 2. Motivation, Related Works and Major Contributions

Recently, vibration monitoring has become a highly reliable condition monitoring paradigm for fluid pumps (and most systems) [11]; consequently, pump manufacturers are recently integrating on-board vibration (and temperature) sensors into their products; however, interpreting and drawing accurate conclusions from the complex big data has been one of the setbacks facing optimized fluid pump FDI and prognostics [6]. As a result that raw vibrational signals are non-stationary with background noise, extracting salient features from these signals remains an on-going challenge for accurately understanding the underlying dynamics of targeted systems. Statistical time-domain, frequency-domain and the more robust time–frequency-domain feature extraction techniques all provide reliable avenues for accurate condition monitoring. However, each technique has its unique pros and cons when representing system dynamics; hence, the need for comprehensive feature extraction and reliable feature selection methods for more accurate system diagnostics and prognostics [3]. A typical study proposed by Pablo et al. [12] presented the effectiveness of statistical time-domain features for multi-fault (11 faults) diagnosis of an electric motor at three gearbox-controlled speeds and loading variations. After extracting 30 statistical features, three feature ranking procedures—ReliefF, Chi-squared and Information Gain tests—were respectively used for selecting the most discriminative features for fault classification using the SVM and the Feed Forward Pattern Recognition Network (FFPN), respectively. The results do not only show the superiority in classification performance of the SVM, but it also shows the salience of the highest-ranking features-mean, CPT5, kurtosis, skewness, etc. for vibration-based diagnosis; nevertheless, against the limitations of the statistical time-domain features to capture spectral information, the time–frequency-domain signal processing techniques for feature extraction becomes necessary.

Inspired by the effectiveness of extracting time–frequency-domain features from acoustic, microseismic and vibrational signals, Mel frequency cepstral coefficients (MFCCs), the 1st-order differential (Delta) and the 2nd-order differential (Delta-Delta) coefficients have

shown effective capabilities for many diagnostics problems [13–16]. For fault diagnosis of bearings and fans, Zhang et al. [13] linearly fused zero-crossing rate (ZCR), MFCCs and Wavelet Packet Decomposition Energy features from acoustic signals from each component and classified each fault/operating condition using the SVM classifier. Results showed higher accuracy than a single MFCC approach. The authors of [14] took a more comprehensive approach by combining (using a vector quantization algorithm) MFCCs with vibrational mode decomposition (VMD) features for detecting common valve clearance fault of diesel engines under different noisy conditions. Classification results using the KNN classifier at variations of diagnostics procedures showed the influence of VMD and the vector quantization for improved diagnostics performance. For fault detection in bearings, Jiang et al. [15] took a more solitary approach by extracting MFCCs and Delta features after a filtering process based on spectral kurtosis. The filtering process was made to extract the non-Gaussian characteristics of the vibration signals from which, these MFCC and Delta features were extracted, and classified using a convolutional neural network, the diagnostics results showed a 98.76% accuracy against seven other existing methods. On a different note, the authors of [16] proposed the use of MFCCs for monitoring the Dongguashan Copper Mine, China using the microseismic events recorded between 13 December 2017 and 17 January 2018. These were classified using a Gaussian Mixture Model-Hidden Markov Model (GMM-HMM) classification model with a 92.46% accuracy. These motivated the use of MFCCs, Delta and Delta-Delta features for our study.

The support vector machines (SVM)'s robustness for fault diagnosis in railway systems [17], bearing diagnosis [13], fault detection in inverter drives [18], etc. is quite remarkable and with its kernel-compatible architecture, has even better diagnostic capabilities; however, for optimum performance, feature selection has become inevitable. Although filter-based methods like the Pearson's correlation, linear discriminant analysis, Chi-squared, etc. as preprocessing techniques help eliminate redundant features by a correlation assessment of features, they cannot be solely relied upon for optimum classifier performance. In contrast, wrapper methods perform optimally since they consider the classifier's performance and aim to retain features that yield the best result. The recursive feature selection for SVM (SVM-RFE) is a reliable and efficient wrapper algorithm whose early success for cancer classification [19] opened doors for many problems [13,17,18]. The RFE uses the classifier's performance as the fitness criteria to score the various subsets of features by ranking the squared weights of each feature; however, unlike the SVM-RFE which computes the feature weights based on a known mapping function, the SVM-RBF has been a "black box" owing to its architecture (due to the nonlinear RBF kernel) and this makes it almost impossible to compute the feature weight vector since the mapping function is unknown. However, based on the superior performance of the SVM-RBF against other SVM kernels on the same input feature set, we were motivated to employ the SVM-RBF-RFE for electromagnetic pump fault diagnosis.

Against the limitations of our earlier sub-optimal approach in [10] whose performance relies significantly on the locally linear embedding (LLE) algorithm and (its parameters), after continued research studies, this study widens the scope by integrating an extra critical failure mode- unspecified power supply into the experiment to ensure that a more reliable and comprehensive diagnostics framework is achieved. In addition, because our past work followed a somewhat suboptimal approach—the use of the 2-dimensional LLE-reduced MFCCs as the input to the SVM* classifier—the efficiency of the past approach would rely immensely on the LLE algorithm and (its parameters) and this has further motivated us to exploring higher-order features with discriminative spectral information, selecting discriminant features by a hybrid feature selection process (correlation and recursive feature elimination (RFE) algorithm), and exploring (and comparing) the proposed SVM-RBF-RFE classification model with other popular diagnostics models. In [20], the authors compared several diagnostics tools including the SVM and provided some intuitive paradigms towards making choices amongst the methods and this has also motivated our study. Although the results therein favored the RF against the others, the deduction

remains questionable considering that (1) the features used for the comparison may be insufficient/inadequate, (2) the study lacks a reliable parameterization paradigm for the classifiers which may result to over-fitting and/or under-fitting issues and (3) the impact of discriminative feature selection was not considered. Consequently, this paper significantly presents the following key contributions:

- Proposal of a cost-efficient MFCC-based FDI technology for electromagnetic pumps based on highly discriminant features.
- Implementation of a rank-based feature selection based on Pearson's correlation and asymptotic significance (recursive feature elimination of irrelevant features) for discriminative feature selection.
- Improvement of the earlier proposed MFCC-LLE-SVM* fault detection method for improved FDI.

## 3. Theoretical Background

In this section. the theoretical background of continuous temperature monitoring, time–frequency-domain feature extraction and the SVM-RBF-RFE for fault isolation are discussed.

### 3.1. Comprehensive Feature Extraction

Feature extraction remains an essential procedure required for accurate and reliable condition assessment of systems from its vibrational (or any other physical sensor-extracted) signals. In the following sub-sections, continuous temperature monitoring and time–frequency-domain feature extraction for condition monitoring are discussed.

### 3.1.1. Continuous Temperature Monitoring

In most control systems, temperature sensors provide reliable and valuable information for continuously monitoring the thermal conditions of the systems. Often, such monitoring is needed for both the system and the environment in which it functions. For electromagnetic devices, increased amperage (caused by excessive voltage, overload, unlubricated components, etc.) usually causes the induction coil heat up (increased temperature). By installing temperature sensors such as infrared thermometers, thermocouples, resistance temperature detectors (RTDs), temperature transducers, thermistors, etc. on these pumps, detection of defined conditions are possible to prevent system damage [21].

Although temperature monitoring serves to provide good visual monitoring as temperature varies across operating conditions, the use of temperature data for diagnostics, is quite unreliable since several faults (or operating conditions) could have the same/similar thermal condition; hence, are inefficient for fault detection as we aim to achieve in this study.

### 3.1.2. Time–Frequency-Domain Feature Extraction

Against the limitations of statistical time-domain features to capture characteristic frequencies in vibrational signals, frequency-domain feature extraction was widely being used to provide vital spectral information for identifying and monitoring the various constituent frequencies of a vibrational signal [22]; however, they lack transient information in the data and frequency information can only be extracted over a complete signal duration [23]. Consequently, to mitigate the drawbacks of these homogeneous techniques, time–frequency-domain feature extraction methods come in handy and with better effectiveness.

Vibrational signals are non-stationary in nature and contain low energy level signals in the presence of dominant noise. This makes it an uphill task to identify and isolate such low energy level fault signals; however, time–frequency-domain signal processing techniques have strong capabilities in decomposing vibrational signals into several energy levels [24–26] and this makes it possible for them to identify and isolate fault signals at various levels. The short-time Fourier transform (STFT), wavelet transform (WT) and the empirical mode decomposition (EMD) are some of the popular techniques for time–frequency-domain feature extraction; however, due to their limitations (STFT is limited by the choice of

window function [24], the EMD has a high sensitivity to noise [25] and WT is unreliable for long-range dependencies [26]), the use of MFCCs and their derivatives as reliable features for diagnostics (and prognostics) problems have recently attracted global attention because of their remarkably significant immunity to noise and robustness in extracting underlying nonlinear characteristics from stationary and non-stationary signals [14–16].

Figure 1 shows the schematic procedure for extracting MFCCs from a signal; however, the stages summarized below provide the stages for their extraction.

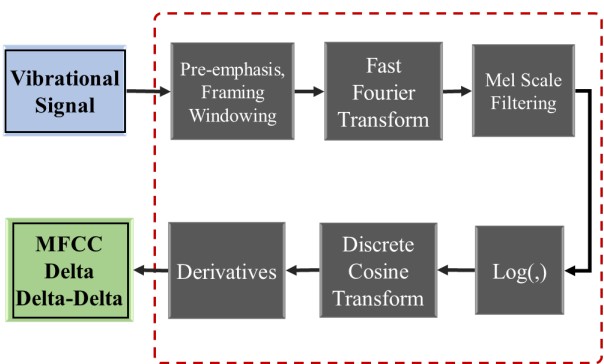

**Figure 1.** Mel frequency cepstral coefficient (MFCC), Delta and Delta-Delta feature extraction processes.

Stage 1: The pre-emphasis, framing and windowing step provides the solution to capturing the spectral information across the non-stationary signals. Since the splitting is done is short frames, each frame is assumed to be stationary. After splitting the vibrational signal into short time frames (preferably between 20–40 milliseconds), compute the fast Fourier transform (FFT) of each of the frames using Equation (1).

$$\overrightarrow{S}(K) = \frac{1}{N} \sum_{i=0}^{N-1} \overrightarrow{A}(t) h(i) e^{-j\left(\frac{2\pi ik}{N}\right)}, 0 \leq i \leq N \tag{1}$$

where $\overrightarrow{A}(t)$ and $\overrightarrow{S}(k)$ are the input time-domain signal and the frequency-domain output of the signal, respectively and $k$ is the length of the FFT. $N$ is the number of frames of the signal and $h(i)$ is the Hamming window whose value depends on a normalization factor ($\beta$).

Stage 2: The frequencies are converted from Hz scale to Mel scale by a process called Mel Warping—square $\overrightarrow{S}(k)$, obtain the energy spectrum and use Mel band filters ($m = 1, 2, \ldots, M$) spaced uniformly on the Mel scale shown in Equation (2) to filter.

$$f(m) = \left(\frac{N}{F_s}\right) B^{-1} \left[ B(f_l) + m \frac{B(f_u) - B(f_l)}{M + 1} \right] \tag{2}$$

where $F_s$ is the sampling frequency, $f_u$ and $f_l$ are the upper and lower frequencies, respectively, $M$ is the number of Mel filters and $B(f)$ and $B^{-1}$ are the Mel scale and its inverse, respectively.

Stage 3: The log energy of each filter bank is computed using Equation (3)

$$\overrightarrow{S}(m) = \ln \left( \sum_{m=0}^{N-1} |\overrightarrow{S}(k)|^2 H_m(k) \right), 0 \leq m \leq M \tag{3}$$

where $H_m$ is the transfer function of the $m$-th filter.

Stage 4: Lastly, to obtain the Mel cepstral coefficients, the logarithmic Mel spectrum is converted back to the time-domain by taking the discrete cosine transform (DCT) of the spectrum using Equation (4)

$$c(n) = \sum_{m=0}^{M-1} \frac{\overrightarrow{S}(m)\cos \pi\theta(m+0.5)}{\Theta} \tag{4}$$

where $\theta$ is the number of frames, and $\Theta$ is the number of MFCCs extracted from $n^{th}$ frame of the signal ($0 \le \theta \le \Theta$).

In practice, the lower order MFCCs (usually 2nd–13th MFCCs) contain more discriminative spectral information from the signal; however, to further extract more discriminative features, the $n$-order difference cepstral coefficients are computed using Equation (5) in addition to the MFCCs to generate a $12(n+1)$-dimensional vector where the first-order differential ($n = 1$) cepstral coefficients constitute of the *Delta* features while the second-order differential ($n = 2$) cepstral coefficients constitute the *Delta − Delta* features.

$$d_i(n) = \frac{\sum_{n=1}^{N} n(c_{i+n} - c_{i-n})}{2\sum_{n=1}^{N} n^2} \tag{5}$$

where $d_i(n)$ is a delta coefficient, from frame $i$ computed in terms of hte static coefficients $c_{i+n}$ to $c_{i-n}$. Basically, at $N = 2$, the acceleration coefficients (*Delta − Delta*) are returned whereas at $n = 1$, the velocity coefficients (*Delta*) are returned.

### 3.2. SVM-RBF-RFE Algorithm for Fault Isolation

Since 1998 [19], the SVM have been found resourceful for many classification problems; hence, its popularity/dominance for systems diagnostics. Being a binary classification algorithm by birth, its potentials for multi-class problems has over the years, been being harnessed and employed for several fault detection/classification problems.

When provided with labeled inputs, this ML technology creates a maximum-margin hyperplane that creates a separation between data points in the same class/label while creating a maximum distance between the classes to the hyperplane(s). This is achieved by a mapping function $\phi$.

For a pair of corresponding vectors in the input space and feature space ($X_i$ and $Z_i$, respectively),

$$Z_i = \phi(X_i) \tag{6}$$

Given a linearly separable set of multi-class data $\{(x_1,y_1),(x_2,y_2),...,(x_n,y_n)\}$, where $x_n \in R^m$ and $y_n \in \{-1,1\}$, a hard margin SVM that separates the data is computed using Equations (7) and (8), respectively:

$$w^T x_i + b \ge 1 \quad \text{for } y_i = 1 \tag{7}$$

$$w^T x_i + b \le -1 \quad \text{for } y_i = -1 \tag{8}$$

where $w^T x_i$ is the weighted training examples, $b$ is the bias, $\xi_i$ ($0 < \xi_i \le 1$) is a softening constraint and $y_i$ is the label.

On the other hand, if the data are not linearly separable as is in this case study, a soft margin SVM which transforms the space of the data to a higher order is quite effective and this is achieved by introducing a softening constraint $\xi_i$ ($0 < \xi_i \le 1$). Consequently, Equations (7) and (8) are updated as Equations (9) and (10), respectively:

$$w^T x_i + b \ge 1 - \xi_i \quad \text{for } y_i = 1 \tag{9}$$

$$w^T x_i + b \le -1 - \xi_i \quad \text{for } y_i = -1 \tag{10}$$

Minimizing the $\|w\|$ increases the distance between the hyperplane; however, its success depends on a regularization parameter, $C$ which controls the relative weight-

ing between ensuring a minimal margin and maximum separation. Consequently, the aim becomes

$$\min \quad \frac{\|w\|^2}{2} + C \sum_{i=1}^{N} \xi_i \tag{11}$$

Employing the Lagrange function in Equation (12) gives the desired output:

$$L(w, b, \alpha, \xi_i) = \frac{\|w\|^2}{2} + C \sum_{i=1}^{N} \xi_i - \sum_{i=1}^{N} \alpha_i \cdot \{y_i \cdot [(w, x_i) + b] - 1\} \tag{12}$$

where $\alpha_i$ is the Lagrange coefficient.

By constraining $\alpha_i : \alpha_i \neq 0 (0 < \alpha_i \leq C)$, the Lagrangian would be bounded so that the optimal hyperplane achieved with the weight vector and the hyperplane function shown in Equations (13) and (14), respectively.

$$w = \sum_{i=1}^{N} \alpha_i \cdot y_i \cdot Z_i \tag{13}$$

$$f(Z) = b + \sum_{i=1}^{N} \alpha_i \cdot y_i \cdot [Z_i \cdot Z] \tag{14}$$

where $[Z_i \cdot Z]$ is the inner product of the two vectors in the input feature space.

Unless with a linear kernel SVM, the weight vector $w$ can be computed directly according to Equation (13) since the mapping function is known. Consequently, employing the RFE algorithm which considers all input feature weights and eliminates features with least squared weight $w^2$ via a backward elimination process, the least ranking features can be eliminated while the features with the most squared weights are considered most discriminant.

On the other hand, when a nonlinear kernel (like the RBF kernel as defined in Equation (15) is employed, the computing the weights becomes a black box since the mapping function is unknown.

$$K(X_i, \overline{X}) = \exp \left( -\frac{|X_i - \overline{X}|^2}{2\sigma^2} \right) \tag{15}$$

where $\sigma$ is the standard deviation and $\overline{X}$ is the mean of $X_i (i = 1, 2, ..., N)$

Some studies have tried solving this problem [24] by expanding the nonlinear RBF kernel into its Maclaurin series, and then computing the weight vector $w$ from the series according to the contribution made to classification hyperplane by each feature; however, the high level of assumptions on which its effectiveness depends. High computational costs associated with this approach makes it non-feasible for near real-time industrial applications. In practice, features selected by the SVM-RFE would perform just as well for the SVM-RBF-RFE classifier with lesser false alarm rate.

Subsequently, the classifier can be constructed using the optimal hyperplane, input data samples and corresponding parameters using the RBF function in Equation (16):

$$f(x) = \text{sign} \left( b + \sum_{i=1}^{N} \alpha_i \cdot y_i \cdot K(X_i, \overline{X}) \right) \tag{16}$$

## 4. Proposed FDI Model

This study aims at developing a robust diagnostic model for FDI of electromagnetic pumps under various running conditions. Figure 2 shows the proposed diagnostics method.

To ensure a comprehensive feature extraction, in addition to MFCC, Delta and Delta-Delta features are also extracted followed by a correlation test using Pearson's correlation ($\rho$), followed by a rank-based selection of uncorrelated features based on their asymptotic

significance using the SVM-RFE. As shown in Figure 2, the Pearson's correlation ($\rho$) test assesses correlation amongst features from which, one amongst highly correlated features is selected while the others are discarded/eliminated. This simple but effective statistical test which was developed by Karl Pearson in 1911 [27] is a measure for quantifying the linear dependence between two variables $X$ and $Y$ and has a range of $-1$ (negative correlation) and $+1$ (positive correlation. This is obtained using Equation (17).

$$\rho_{X,Y} = \frac{\mathrm{cov}(X,Y)}{\sigma_X \sigma_Y} \tag{17}$$

where $\sigma_X$ and $\sigma_Y$ are the standard deviations of $X$ and $Y$, respectively while $\mathrm{cov}(X,Y)$ is the covariance.

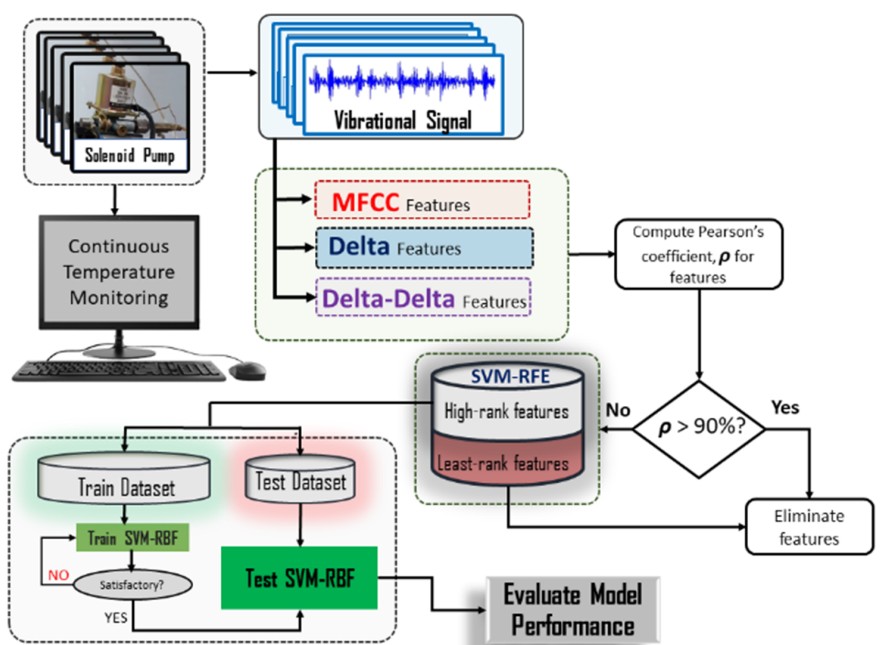

**Figure 2.** Proposed fault detection and isolation (FDI) model.

In an unsupervised manner, each feature's correlation between the other features are simultaneously computed using Equation (17). Negatively correlated (uncorrelated) features are retained while features with a correlation value greater than (or equal to) a threshold value of 0.9 are eliminated. This is to ensure that only the highly correlated features are eliminated. Apart from dimensionality reduction, this pre-processing step easily eliminates the chances of classifier confusion—a situation whereby non-discriminant/redundant features are used as inputs for a classifier.

Next, using the uncorrelated feature set as input, the SVM-RFE is used to select the most important features (high-ranking features) while the least-ranking features are eliminated. When the SVM accepts the input features, it assigns weights to them. These weights then constitute the criteria for feature importance. The RFE's goal is to recursively assess these features based on their respective weights. In other words, the SVM is internally wrapped by the RFE for highly discriminant feature selection.

Features extracted from a portion of vibration signals are finally used to train the SVM-RBF classifier while the remaining data files are used for testing. Upon achieving optimum training by tuning the $C$ parameter and selecting the appropriate RBF gamma ($\gamma$) parameter, model performance evaluation is done using standard classification performance metrics: precision, sensitivity, F1-score, false positive rate and test accuracy [26].

## 5. Experimental Analysis

This section presents the experimental validation of the proposed FDI methodology on five VSC63A5 electromagnetic pumps running under the conditions summarized in Table 1.

All experiments were carried out at room temperature at the Defense Reliability Laboratory, Kumoh National Institute of Technology (KIT), Republic of Korea.

**Table 1.** Pump working conditions.

| Pump Label | Input Power | Working Fluid Composition | Failure Mode | Class Label |
|---|---|---|---|---|
| VSC-1-CNT | 220 V, 60 Hz | 4 L Diesel, 5 g Paper Ash | Contamination Fluid | 0 |
| VSC-2-VISC | 220 V, 60 Hz | 3 L Diesel, 3 L SAE40 Engine Oil | Highly Viscous Fluid | 1 |
| VSC-3-CLOG | 220 V, 60 Hz | 4 L Diesel, 1 g Paper Ash, 0.2 L Paraffin Solution, 100 g Pectin Powder | Clogged Suction Filter | 2 |
| VSC-4-NORM | 220 V, 60 Hz | 10 L Clean Diesel | Normal | 3 |
| VSC-5-AMP | 300 V, 40 Hz | 10 L Clean Diesel | Unspecified power supply | 4 |

### 5.1. Experimental Setup

Our test-bed shown in Figure 3 consists of five VSC63A5 electromagnetic pumps with a standard rating of (220 V, 60 Hz) at various operating conditions. To simulate these fault conditions, paper ash (after being burnt) was sieved with an 88 microns sieve. Then, 5 g of sand fractions were mixed with 4 L of diesel and the mixture used as the working fluid for VSC-1-CNT. Equal volumes of SAE40 engine oil and diesel were mixed to simulate a relatively highly viscous fluid for VSC-2-VISC while 1 g of the gravel fraction from the paper ash was measured and mixed with 100 g of pectin powder and 0.2 L of paraffin solution and prepared for use by VSC-3-CLOG. The mixture was stirred continuously by an overhead stirrer (OSA-10 made by LK LABKOREA) to ensure a Brownian motion of the contaminants (and minor precipitates) in the working fluid. After the experiment, these particles and precipitates clogged the suction filter in VSC-3-CLOG. For VSC-4-NORM and VSC-5-AMP, clean diesel fuel was prepared for each of the pumps.

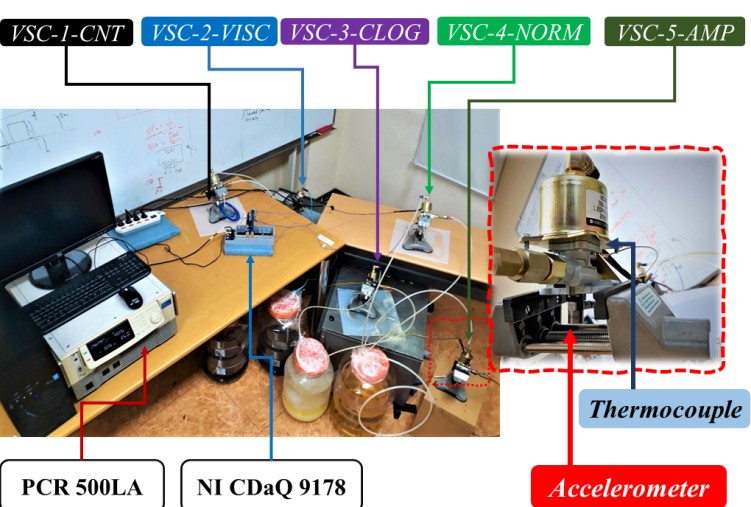

**Figure 3.** Photo of experimental setup.

Apart from VSC-5-AMP, which was powered by a PCR 500LA variable AC power supply to simulate an unspecified input power condition, the other pumps were powered by a 220 V, 60 Hz automatic voltage regulator (AVR). In addition, the suction filter was removed in all the pumps except VSC-3-CLOG to avoid filter clogging and ensure a free run of the respective fluids through the pumps. The pumps were operated under the recorded conditions for 10 days (8 h per day) while being monitored. These recorded working conditions were chosen due to recent findings on these conditions being the most probable failure modes these pumps (and solenoid valves) are prone to [6].

Through a NI 9234 module, the vibrational signals were stored via high sensitivity accelerometers attached vertically under each pump's rotor casing as illustrated in Figure 3 (Right). In addition, the real-time temperature data of the pumps were stored for visual monitoring via a NI 9212 module through a two-wire thermocouple attached as also shown in Figure 3 (Right).

Both modules were connected to an NI CompactDAQ 9178 and through a LabVIEW interface, the signals were sampled at 1 KHz and saved in a *.csv* file format.

## 5.2. Experimental Results

At the end of the experiment, the following were observed: A contamination in the pressure chamber of VSC-1-CNT, an observable stress-induced vibration by VSC-2-VISC, a clogged suction filter in VSC-3-CLOG and a coil burn-out in VSC-5-AMP. Apart from VSC-4-NORM which operated as healthily as expected, VSC-1-CNT was observed to heat more than the other pumps (as seen from Figure 4b). This was also accompanied by a reduction in pump pressure due to cavitation. VSC-2-VISC also showed a significant reduction in pressure (accompanied by rumbling sounds) resulting from the pump's effort to pump a more viscous/heavier fluid. VSC-3-CLOG, whose suction filter was clogged, showed an increase in vibration (with rumbling sound) with a reasonable amount of cavitation and reduced output pressure while VSC-5-AMP showed a reduced output pressure and an increased vibration with a crackling sound owing to increased amperage and overload on the solenoid/electromagnetic coil.

For each operating condition, 100 min of vibrational data were collected in 10 data files and labeled as shown in Table 1. Figure 4a shows a view of the raw vibration signals of the pumps while Figure 4b shows their varying continuous temperature monitoring results (used for visual monitoring only) for the first few minutes of operation.

**(a)**

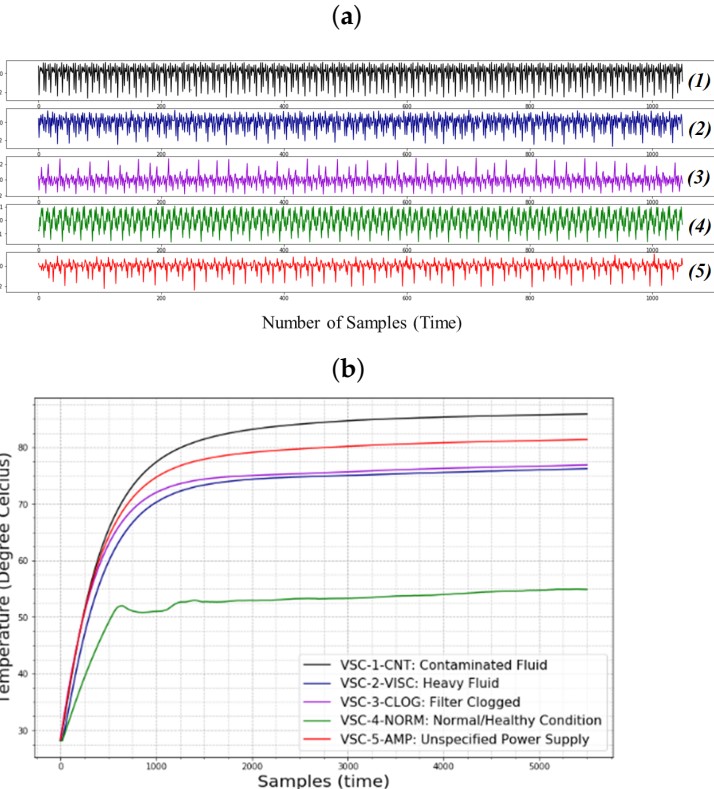

**(b)**

**Figure 4.** (**a**) Raw vibrational signals and (**b**) continuous temperature monitoring results for (1) VSC-1-CNT: Contaminated Fluid, (2) VSC-2-VISC: Heavy Fluid, (3) VSC-3-CLOG: Filter Clogging, (4) VSC-4-NORM: Normal/Healthy Condition and (5) VSC-5-AMP: Unspecified Power Supply.

### 5.3. Comprehensive Feature Extraction and Selection

From each pump's vibrational signals, I2 MFCC, 12 Delta and 12 Delta-Delta features (2nd to 13th MFCCs, Deltas, and Delta-Deltas) were extracted respectively while the rest were discarded because the higher DCT coefficients reflect fast changes in the filter bank energies and as verified in [13,14], these fast changes affect classification performance. Subsequently, a 36-dimensional feature vector was produced and prepared for feature selection.

The extracted features were checked for a high positive correlation (one feature amongst a set of features with a correlation above 90% is selected while the rest are dropped). After the correlation test, 34 uncorrelated features were obtained while 3rd and 4th Delta features were dropped due to their high correlation with 2nd Delta. Next, the 34 uncorrelated features were used to train an SVM-RFE model to obtain the 20 most discriminant features. Figures A1–A3 (Appendix A) present the correlation plots between the features from start to the end of the feature selection process. As shown in Figure A3, the 20 high-ranking features have very high discriminance (low correlation) amongst them. Figure 5 shows the ranking performance of the 34 uncorrelated features.

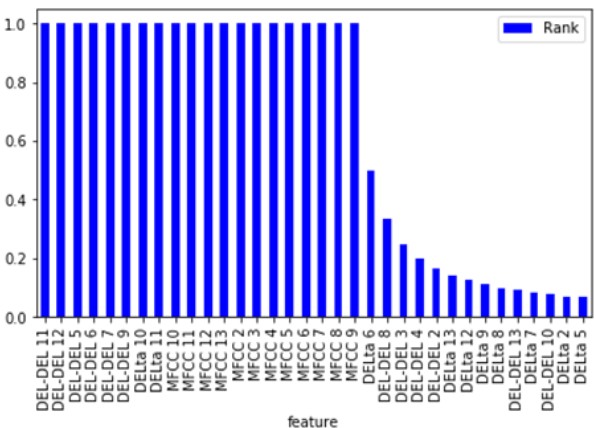

**Figure 5.** SVM-recursive feature elimination (RFE) ranking performance of the 34 uncorrelated features.

As shown, the features with the most squared weights are assigned a rank of value 1 while the features with least squared wights rank between 0 and 1.

### 5.4. Fault Isolation by the SVM-RBF-RFE

First, the 20 high-rank features from the training data set were used to train the SVM-RBF model. To achieve this, about 70% of the whole data files are used for training while the remainder (about 30% of the data files) were used for testing.

Importantly, the choice of SVM parameters greatly affects its classification capability. On a more practical sense, the RBF kernel, generally has more isolation capabilities for multi-class problems. As a result, its practical use depends hugely on the combination of the $C$ and $\gamma$ parameters and determines the robustness of the approach against other methods. Large $C$ and $\gamma$ parameters lead to over-fitting and prolong time for training while the reverse is the case for small $C$ and $\gamma$ values; therefore, choosing an optimal combination of these parameters in the most non-exhaustive manner becomes necessary for cost-efficient industrial applications. This problem can be mitigated by a grid search for an optimal combination amongst an exponential range of parameters (like $C$, $\gamma \in \{10^0, 10^1, 10^2\}$) [28]. This is because these set of parameter combinations has proven records for reducing over-fitting. Subsequently, a grid search on 10 SVM-RBF models consisting of $C, \gamma \in \{10^0, 10^1, 10^2\}$ combinations was done over a 10-fold cross-validation respectively, with the SVM-RBF[$C = 10, \gamma = 1$] returning the most accurate classification report (99.46% test accuracy).

To further verify the claim that features selected by the SVM-RFE would perform just as well for the SVM-RBF-RFE classifier and present a visual classification result, a standard locally linear embedding (LLE) ($NN = 15$) algorithm [10] was employed for dimensionality reduction of the 20 high ranking features to a 3-dimensional feature set for visualization in 3D space. The selected features are first normalized (between 0 and 1) and the first three local embeddings (LLE 1, LLE 2 and LLE 3) of the features are obtained linearly. Figure 6 shows the isolation results in 3D where the colors light blue, dark blue, green, yellow and red represent VC-1-CNT, VSC-2-VISC, VSC-3-CLOG, VSC-4-NORM and VSC-5-AMP, respectively. Figure 7 shows the isolation plot (in 2D) of the SVM-RFE and SVM-RBF-RFE classifiers with identical parameters ($C = 10$, $\gamma = 1$), respectively. The horizontal and vertical axes represent the first and second local embeddings (LLE 1 and LLE 2), respectively while the color assignment for the pumps are retained as in Figure 6.

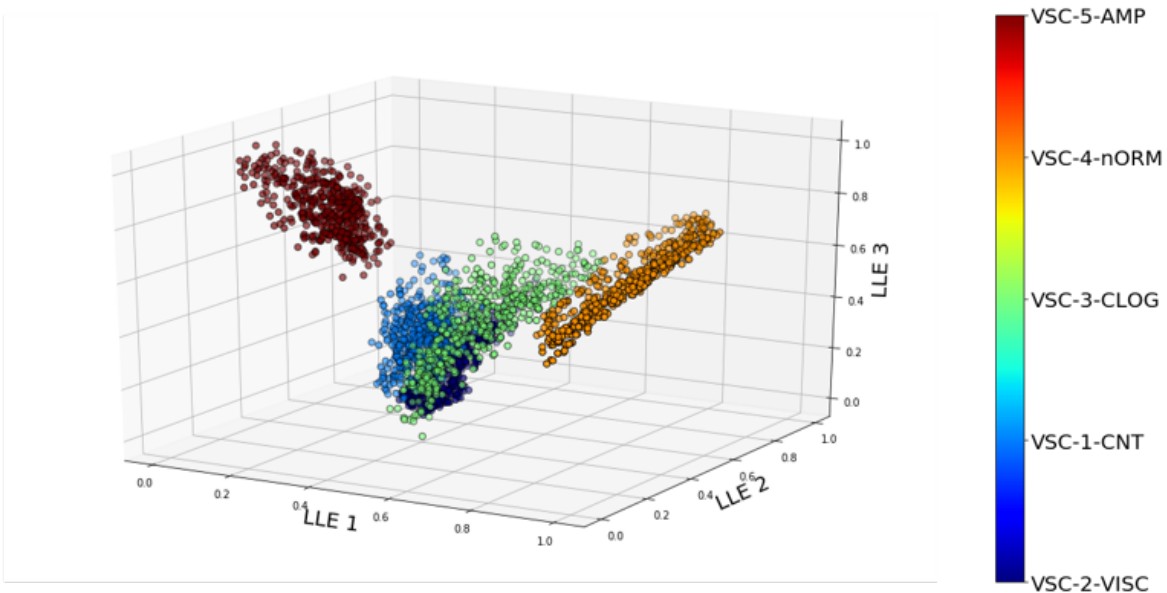

**Figure 6.** 3D-view of the locally linear embedding (LLE)-transformed features showing high discriminance ($NN = 15$).

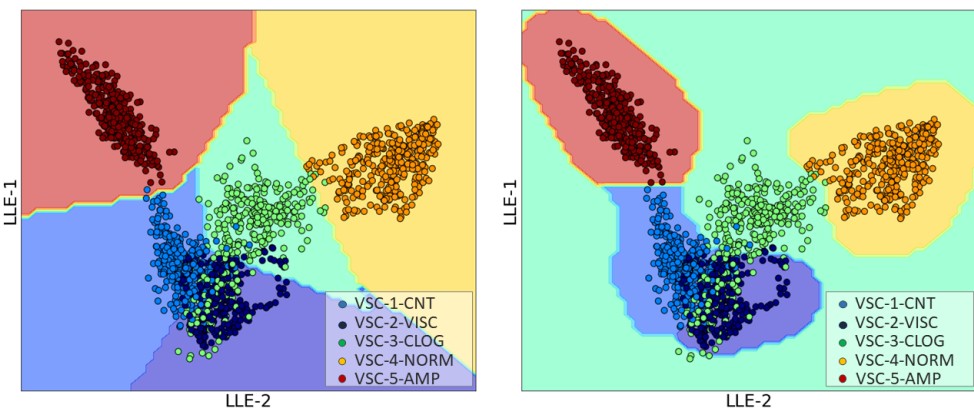

**Figure 7.** Fault isolation results of high-ranking features with (**Left**) showing classification performance of SVM-RFE and (**Right**) showing SVM-RBF with RFE classification results.

As shown, the transformed features are separable (due to high discriminance of the selected features) while the SVM-RBF-RFE classifier shows a better isolation performance on the LLE-transformed features ($NN = 15$). This provides reliable intuition on the better isolation performance of the SVM-RBF-RFE against the conventional alternative.

### 5.5. Performance Evaluation and Comparison

The analysis herein aims at addressing the aforementioned challenges in [10,20] by exploring the efficiency of the SVM-RFE on deeper spectral features. This shall not only ensure that optimal FDI performance is achieved using the salient features at lesser computational costs, but the strenuous manual parameter tuning stage associated with other methods can be avoided by employing the grid search technique over a range of SVM parameters.

To assess the performance of the proposed FDI technology, standard classifier performance evaluation criteria were employed for performance evaluation of the proposed model alongside computational costs associated with its implementation. All analyses were done using Python 3.7 on a desktop computer with the specifications: AMD Ryzen 7 (manufactured in Taiwan), 2700 Eight-core 3.20 GHz processor and 16 GB RAM. It is important to note that computation costs (speed) depend hugely on the computer configuration whereby a high-speed, large memory computer configuration returns a faster implementation process and vice versa.

First, Table 2 shows the classification results of the SVM-RBF[$C = 10$, $\gamma = 1$] on the test data while Figure 8 shows the confusion matrix and receiver operating characteristic (ROC) curve of the model's performance.

**Table 2.** Proposed model classification results.

| Pump Label | Precision | Recall | F1-Score |
|:---:|:---:|:---:|:---:|
| VSC-1-CNT | 0.95 | 0.98 | 0.97 |
| VSC-2-VISC | 1.00 | 1.00 | 1.00 |
| VSC-3-CLOG | 0.98 | 0.95 | 0.96 |
| VSC-4-NORM | 1.00 | 1.00 | 1.00 |
| VSC-5-AMP | 1.00 | 1.00 | 1.00 |

vspace-12pt As shown in Figure 8a, the model returns zero (0) false positives (FP) and false negatives (FN) for VSC-2-VISC (*Class 1*), VSC-4-NORM (*Class 3*) and VSC-5-AMP (*Class 4*). In contrast, it appears VSC-1-CNT (*Class 0*) and VSC-3-CLOG (*Class 2*) returned the highest FP and FN, respectively with four (4) wrongly classified samples. Invariably, VSC-1-CNT returned a FN equal to the FP of VSC-3-CLOG each with two (2) wrongly classified samples. This is further reflected in Figure 8b which shows the lower area under the curve (AUC). As much as the model was able to all the classes (Pumps) accurately, while VSC-1-CNT and VSC-3-CLOG appear to have smaller AUCs (shown in blue and red colors, respectively); nevertheless, we further compared the proposed model with ten (10) standard classification algorithms, namely: SVM-RFE, quadratic discriminant analysis (QDA), AdaBoost classifier (ABC), gradient boosting classifier (GBC), naive Bayes classifier (NBC), K-nearest neighbor (KNN), support vector machines (SVM), multi-layer perceptron (MLP), neural network, random forest (RF), decision tree (DT) and the traditional logistic regression (LR).

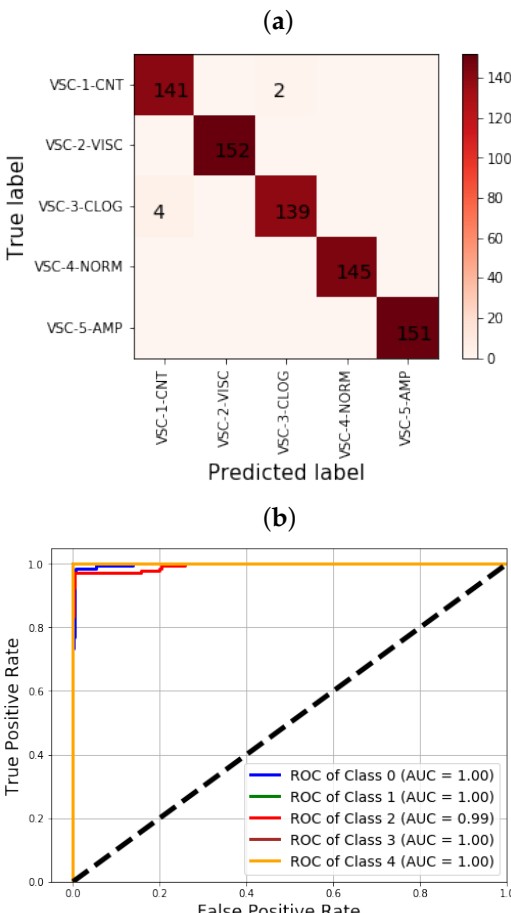

(a)

(b)

**Figure 8.** SVM-RBF with RFE performance results showing (**a**) confusion matrix and (**b**) ROC curve.

Figure 9 shows the comparison in FDI performance of the classifiers where the blue dotted lines with squared intersects represent their respective test accuracies while the red dotted lines with round intersects represent their respective training accuracies. The purple dotted lines with triangle intersects represent their respective training times (in percentage of the total training time of the whole classifier).

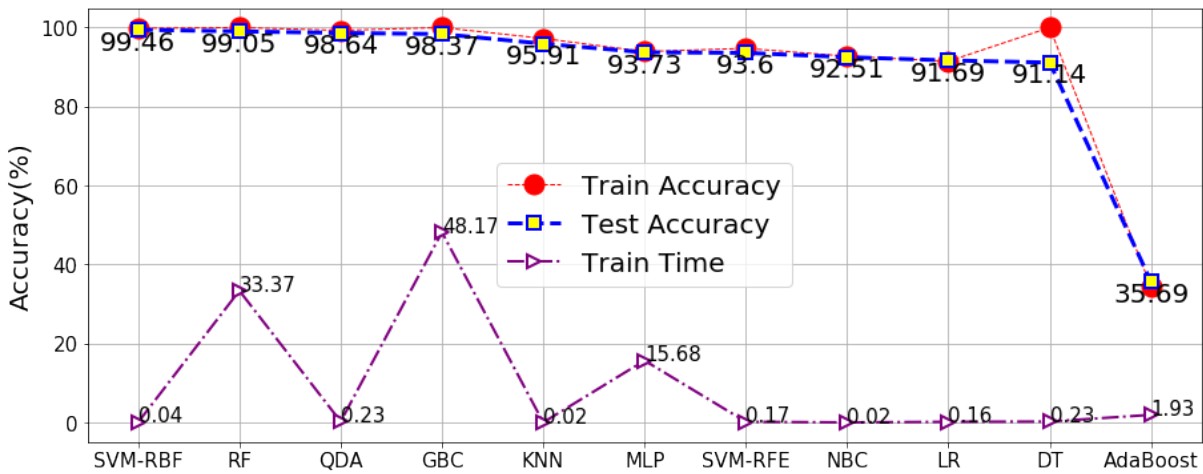

**Figure 9.** SVM-RBF-RFE isolation performance comparison with other classifiers.

As shown, the SVM-RBF-RFE outperformed these other models based on test accuracy; however, when considering the computation cost, although not the most accurate (a test

accuracy of 99.46%), the KNN and NBC were the fastest (with a train time of 0.02% each) while in contrast, the GBC and RF are the most computationally expensive (even though they both rank amongst the most accurate). Nevertheless, with a train time of 0.04% and the most accurate with the least false alarm rate, the SVM-RBF-RFE model is quite computationally cost-efficient.

### 5.6. Cost Analysis of the Proposed FDI Method

As proposed in this study, the major contribution of this experimental study is to verify the computational cost-efficiency of the proposed FDI method. Already, some argue the cost-inefficiency of the SVM as a FDI tool [20]; however, as this study shows, the SVM functions quite optimally in a very computationally friendly manner when provided with the right inputs—highly discriminant features—and with the right architecture, not only will a minimal false alarm rate be achieved, its use for near-real-time applications can be relied upon. This subsection compares the SVM-RBF-RFE's efficiency with several scenarios to assess its cost-efficiency in real situations.

Using the same SBM-RBF model with the same parameters, we compared the model's performance on the following cases which correspond to the number of feature whose correlation map is shown in Figures A1–A3, respectively:

- *Scenario*1: The use of 20 highest-ranking features.
- *Scenario*2: The use of all 34 uncorrelated features.
- *Scenario*3: The use of all 36 extracted features.

Figure 10 presents computational cost (expressed in training time) and accuracy comparison for the three scenarios.

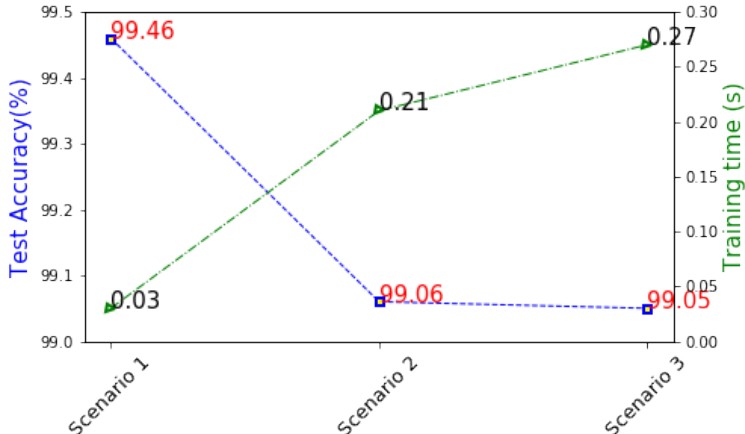

**Figure 10.** Computational cost (training time) and accuracy comparison of the SVM-RBF-RFE in different feature set scenarios.

As shown, an inverse relationship is seen between the test accuracy and computational cost with *Scenario*1 returning the most accurate FDI performance (test accuracy of 99.46%) at the lowest computational cost (train time of 0.03 s) with the least, but highly discriminant number of features. This is in sharp contrast to Scenario 2 and Scenario 3 which returned a lesser and the least test accuracy with a higher and the highest computational costs, respectively. Although the decline in accuracy is shown between Scenario 1 and the other scenarios, the impact of highly correlated features (Scenario 3) does not affect the accuracy as much (only a difference by 0.01% is observed) because they are just a few (Delta 3 and Delta 4). However, the difference in computational costs of Scenario 2 and Scenario 3, in comparison with Scenario 1 shows the advantage of salient/discriminative feature selection for reduced computational costs. This validates the impact of highly discriminant features for improved cost advantage and diagnostics accuracy—the major contributions of this study.

## 6. Conclusions

As an extended version of our previous study, this paper presents a cost–efficient and reliable vibration–based fault detection and isolation (FDI) technology for VSC63A5 electromagnetic pumps based on SVM-RBF-RFE and discriminative features. By extracting Mel frequency cepstral coefficients (MFCCs) and their $n$–order differentials, highly reliable feature extraction was achieved; however, each feature has its uniqueness and weakness for fault isolation. Selecting discriminant features via a hybrid feature selection process (correlation and recursive feature elimination (RFE) algorithm) was proposed. Subsequently, a rank-based feature selection based on recursive feature elimination of uncorrelated features was implemented for salient feature extraction.

With these features, a Gaussian kernel support vector machine was employed for isolation. In comparison with other classifiers, results show that the proposed feature selection approach computationally cheaper and significantly improves diagnostics performance. The accuracy and cost efficiency of the proposed FDI methodology was further assessed in three different scenarios which consists of various feature-sets. Results show that the proposed FDI technology is not only computationally efficient but also yields highly accurate FDI results with minimal false alarm rate.

The proposed scheme can be enhanced to account for other failure modes which may form the scope of building a prognostics scheme for the pumps. Since the faults were artificially designed (against the natural fault behavior in real life situations), real-time applications may be limited so future research studies would be aimed at investigating possible solutions for real-time applications.

**Author Contributions:** Conceptualization, U.E.A.; methodology, U.E.A.; software, U.E.A.; formal analysis, U.E.A.; investigation, U.E.A.; resources, U.E.A. and J.-W.H.; data curation, U.E.A.; writing—original draft, U.E.A., writing—review and editing, U.E.A. and J.-W.H.; visualization, U.E.A.; supervision, J.-W.H.; project administration, J.-W.H.; funding acquisition, J.-W.H. Both authors have read and agreed to the published version of the manuscript.

**Funding:** This research was supported by the MSIT (Ministry of Science and ICT), Korea, under the Grand Information Technology Research Center support program (IITP-2020-2020-0-01612) supervised by the IITP (Institute for Information & communications Technology Planning & Evaluation).

**Data Availability Statement:** The data presented in this study are available on request from the corresponding author. The data are not publicly available due to laboratory regulations.

**Acknowledgments:** This research was supported by Seolha Kim affiliated with Intelligent Robotics Laboratory, Kumoh National Institute of Technology. Her expertise in LabVIEW programming helped for designing the data collection interface for vibration and temperature data collection and storage.

**Conflicts of Interest:** The authors declare no conflict of interest.

## Abbreviations

The following abbreviations are used in this manuscript:

| | |
|---|---|
| ABC | Ada-boost Classifier |
| AI | Artificial Intelligence |
| AVR | Automatic Voltage Regulator |
| DAQ | Data Acquisition |
| DCT | Discrete Cosine Transform |
| DL | Deep Learning |
| DT | Decision Tree |
| EMD | Empirical Mode Decomposition |
| EOL | End-of-life |
| FDI | Fault Detection and Isolation |

| FFPN | Feed Forward Pattern Recognition Network |
| --- | --- |
| FFT | fast Fourier Transform |
| GBC | Gradient Boosting classifier |
| GMM-HMM | Gaussian Mixture Model-Hidden Markov Model |
| NI | National Instrument |
| NN | Nearest Neighbor |
| KNN | K-Nearest Neighbor |
| LLE | Locally Linear Embedding |
| LR | Logistic Regression |
| MFCC | Mel Frequency Cepstral Coefficient |
| ML | Machine Learning |
| MLP | Multi-Layer Perceptron |
| NBC | Naive Bayes Classifier |
| PHM | Prognostics and health management |
| QDA | Quadratic discriminant analysis |
| RBF | Radial Basis Function |
| RF | Random Forest |
| RFE | recursive feature elimination |
| RTD | Resistance Temperature Detector |
| SVM | support vector machine |
| SVM-RBF | Gaussian kernel SVM |
| SVM-RBF-RFE | Gaussian kernel SVM with RFE |
| SVM-RFE | SVM with RFE |
| STFT | short-time Fourier transform |
| VMD | vibrational mode decomposition |
| WT | Wavelet Transform |
| ZCR | Zero-crossing rate |

## Appendix A. Feature Selection Results

This document presents feature selection results from the initial 36–dimesnional feature set (12 MFCCs, 12 Delta coefficients, and 12 Delta-Delta coefficients) to the optimal 20–dimensional feature set. The aim is to show the discriminance of the selected features in comparison with the full feature set using correlation plots. As shown in Figure A1, the Delta 3 and Delta 4 coefficients are highly correlated with the Delta 2 coefficient (with a *p*-value of 0.72 respectively).

Following the first feature selection step– a correlation test with a threshold of 0.7, the 3rd and 4th Delta coefficients were eliminated leaving the 34–dimensional feature set whose correlation plot is shown in Figure A2. This filter-based approach ensures that as much as possible, the full input variables characteristics are well assessed from a statistical viewpoint thereby ensuring that the inter-relationships/dependence of the input variables are well assessed/compared.

- The cost of employing the RFE algorithm on the full feature set is reduced,
- Features for prognostics could be easily detected. The highly correlated features which although are irrelevant for the proposed diagnostics scheme, can be highly reliable for prognostics purposes, and
- The overall computational cost reduced.

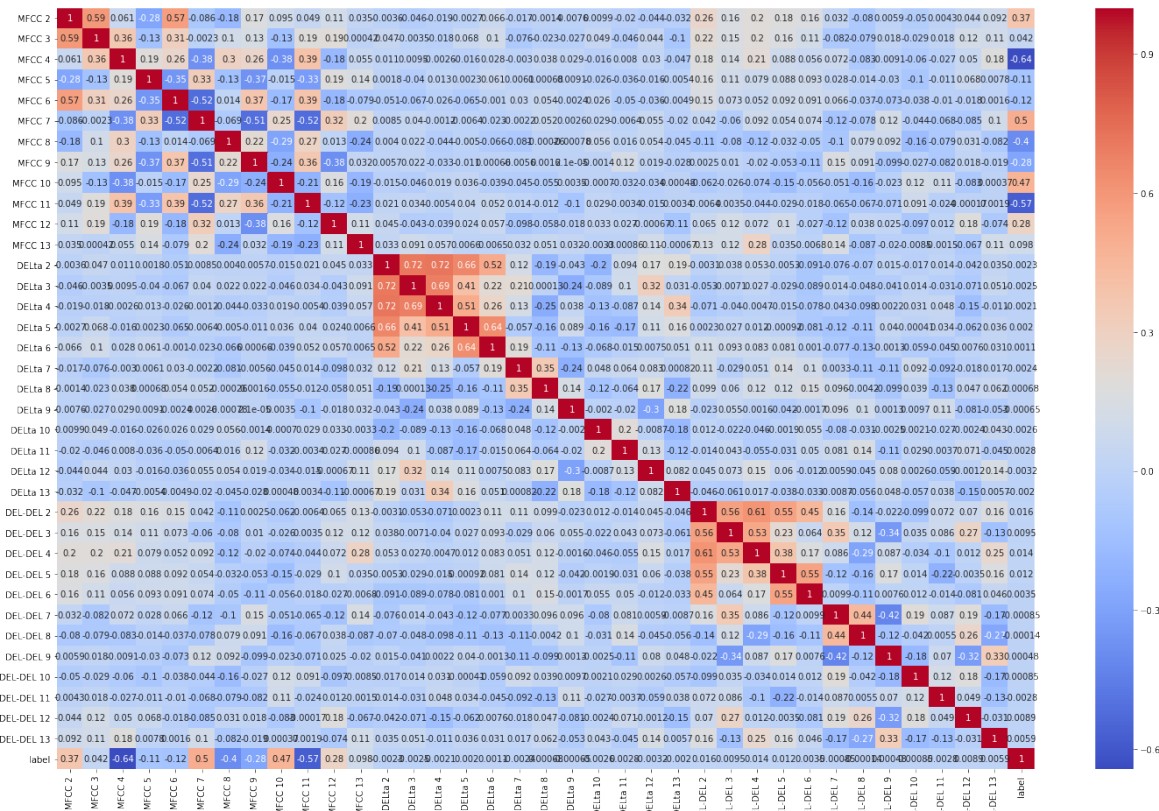

**Figure A1.** Correlation matrix of all the 36–dimensional feature vector.

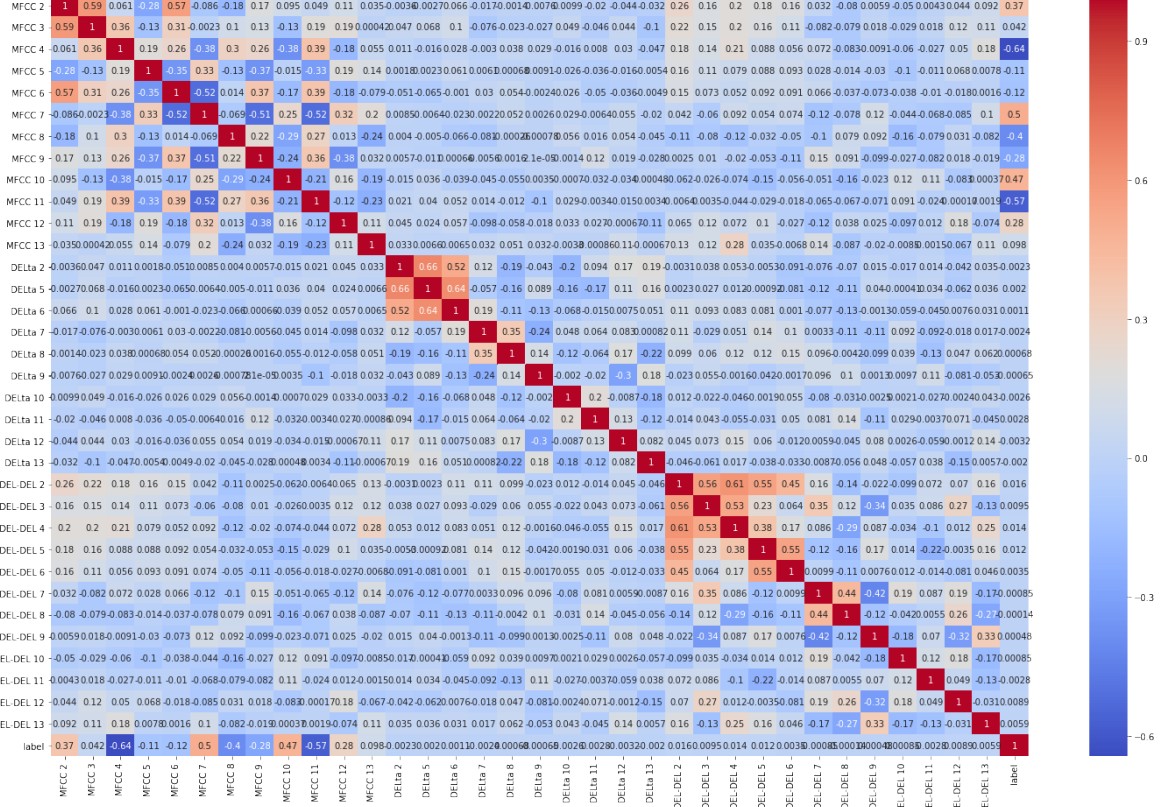

**Figure A2.** Correlation matrix of the 34–dimensional feature vector obtained by the filter–based feature selection process (Pearson's correlation test).

Lastly, Figure A3 presents the correlation plot of the 20 highest ranking features and as shown, all the selected features are very uncorrelated with a maximum *p*–value of 0.59 existing between the MFCC 2 and MFCC 3.

By following multiple feature assessment viewpoints (statistical and RFE–based), h salient features can be selected based on discriminance levels/ranking which minimizes the curse of dimensionality and classifier confusion, thereby minimizing computational costs. This is the bedrock and motivation for this study.

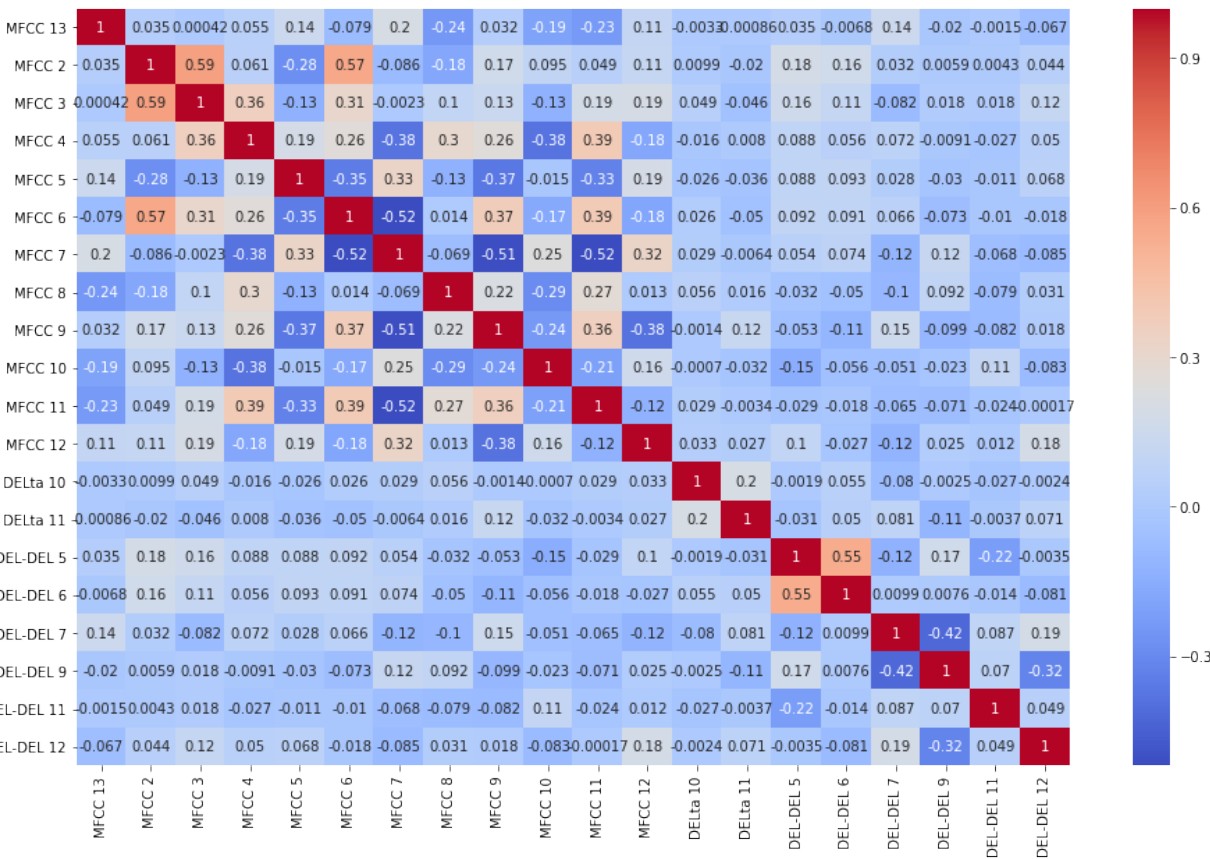

**Figure A3.** Correlation matrix of the 20–dimensional highest–ranking feature vector obtained by the SVM-RFE algorithm.

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
