# Peer review of "A Cost-Efficient MFCC-Based Fault Detection and Isolation Technology for Electromagnetic Pumps"

_electronics, doi:10.3390/electronics10040439_

Round 1
Reviewer 1 Report
The need for proper condition monitoring and health assessment of the hydraulic pumps cannot be overemphasized and this has resulted in extensive research studies on standard techniques for ensuring optimum fault detection and isolation (FDI) results of these pumps. Interestingly, mechanical vibrational signals reflect operating conditions and by exploring the robust time-frequency-domain feature extraction technique(s), the nonlinear system characteristics can be captured for reliable fault diagnosis/condition assessment.
This study is based on the use of vibrational signals for fault isolation of electromagnetic pumps. From the vibrational signals, Mel frequency cepstral coefficients, the first-order and the second-order differentials were extracted and the salient features selected by a rank-based recursive feature elimination (RFE) of uncorrelated features. The proposed model was tested and validated on five VSC63A5 electromagnetic pumps at various fault conditions and isolated/classified using the Gaussian kernel SVM (SVM-RBF-RFE). Results show that the proposed feature selection approach is computationally cheaper and significantly improves diagnostics performance over SVM-RBF-RFE models.
This is an interesting paper with hardware results. However, it appears to be an incremental contribution over the state of the art as mentioned by the author’s themselves. Nevertheless the authors have to highlight the novel features of their work over the others and why their effort is not incremental given extensive results availability. Also, theu authors have to be explain why their work renders better results over others. The false positive and negative aspect of the study need to be included.
Reviewer 2 Report
In my opinion, the article is interesting. Describes the problem of selecting the characteristics of the vibration signal, comparing different methods.
However, Authors should check spelling and punctuation, as I have noticed minor errors such as re-liable (Line 108), pa-per (Line 118), extract-ed (Line 245), cor-relation (Line (298). The text also includes redundancy of words such as: of of (Line 27), have have (Line 28), with with (Line 415), for for (Line 424).
Author Response
Point 1: In my opinion, the article is interesting. Describes the problem of selecting the characteristics of the vibration signal, comparing different methods.
However, Authors should check spelling and punctuation, as I have noticed minor errors such as re-liable (Line 108), pa-per (Line 118), extract-ed (Line 245), cor-relation (Line (298). The text also includes redundancy of words such as: of of (Line 27), have have (Line 28), with with (Line 415), for for (Line 424).
Response 1: We appreciate the reviewer for his positive comments on our manuscript. We are glad that our manuscript meets his satisfaction and recommendation for publication.
As rightly pointed out by the reviewer, there are some spelling errors and we have carefully corrected them all in the updated manuscript and improved its readability.
Reviewer 3 Report
This paper develops an approach for fault isolation of electromagnetic pumps based on the use of vibrational signals. From the vibrational signals different coefficients are extracted, selected and ranked by applying a rank-based recursive feature elimination algorithm. The study is interesting, and experimental results are presented (five VSC63A5 electromagnetic pumps at various fault conditions), which allow validating the proposed approach. The authors should answer the following remarks in order to improve some aspects of the manuscript,
- Section 2. The novelties, contributions and advantages of this paper with respect the state of the art and with respect to reference [7] must be further highlighted and stressed.
- The approach based in this problem highly relies on the FFT. However, it is well known that this transform is only suitable for stationary signals. Current pumps driven by variable frequency drives tend to operate under non-stationary conditions. The authors must acknowledge this aspect and propose solutions.
- Table I. The failure modes described here must be described as well as the means used to generate such faults.
- (17) describes the Pearson correlation coefficient, which quantifies the linear dependence between two variables. Since is calculated using two variables each time, it is not clear how they are ranked in descendent order as in Fig. 2 or if they are ranked by the SVM-RFE algorithm. Please provide more details.
- Section 5.4. “First, the 20 high-rank features were finally used for the train/test process by the SVM-RBF model. To achieve this, 70% of the features chosen randomly for training while the remainder were used for testing throughout the whole analysis.” This is not the way classification problems are solved. The usual approach is to have a training set composed of many signals of each fault type and a test set also composed of many signals of each fault type. It is difficult to understand how the authors solve this classification problem using only one signal of each fault type.
- The origin of the codes used (Matlab generic, own code, etc.) must be specified.
I hope a review of the paper based on these remarks can help to improve the paper quality.
Reviewer 4 Report
The paper tackles issues of fault detection and isolation quality for electromagnetic pumps. The method based on the mel-frequency cepstral coefficient is proposed. The paper might have a value for science and the general science, however the quality of communication have to be improved before publication:
The abbreviation MFCC in the title should be provided at the beginning of the paper, e.g. in Abstract in line 9, not in line 80.
Lump citations, e.g. line 83 should be avoided.
In all equations please mark vectors as bold or with an appropriate arrow sign above the symbols.
Line 34 – Naive
Please unify the style of markings’ presentation – bold or not, italic or not, e.g. line 337 vs. line 341.
Please correct punctuation in the whole text, e.g. line 334, 344 etc.
Eq. (1) -is j^2=-1 or marking j is just an index (please see. Eq. 15). Please clarify and do not repeat symbols.
Eq. (2) – f[m] -> f(m).
Eq. (2) please describe fl and fu
Eq. (4) – slash -> horizontal division line.
Eq. (5) – please define Theta function.
Eq. (16) – please define signum function.
Fig. 5 – what is on this figure, please cite it before appearance (not in line 304). Define rank.
Fig. 6 – please define LEE1, LEE2, LEE3 and the markings on the scale.
Fig. 7 – describe axis and legends.
Figs. 7, 9, 10 - do not finish the section with a figure. Please provide an additional discussion.
Fig. 8 – please describe all classes in the text.
What does “train” mean?
Please update the list of Abbreviations. A number of markings used in the text are missing.
Round 2
Reviewer 3 Report
The authors have replied all my questions
Author Response
We appreciate the reviewer for his positive comments on our manuscript. We are glad that our manuscript meets his satisfaction and recommendation for publication.
Reviewer 4 Report
The authors addressed most of my remarks. The paper is better now. Thank you. However there are some issues mentioned in my previous reviews which are still valid and should be addressed before publication. They are minor remarks but might improve the quality of the paper.
Point 4: Line 406 – Naïve -> Naive
Point 5 – Please check punctuation carefully, e.g., line 392 – 16GB -> 16 GB, line 406 – Classifier(ABC) -> Classifier (ABC), line 407 Neighbor(KNN) -> Neighbor (KNN) etc.
Point 7 – please do not mix markings. If “j” is an imaginary unit then do not use it as an index. Just use another letter as index.
Point 16 – do not finish a section with an object, e.g. Fig. 5, Table 1 etc. Provide an additional discussion after an object.
